# Evaluating AI Methods for Pulse Oximetry: Performance, Clinical Accuracy, and Comprehensive Bias Analysis

**DOI:** 10.3390/bioengineering11111061

**Published:** 2024-10-24

**Authors:** Ana María Cabanas, Nicolás Sáez, Patricio O. Collao-Caiconte, Pilar Martín-Escudero, Josué Pagán, Elena Jiménez-Herranz, José L. Ayala

**Affiliations:** 1Departamento de Física, FACI, Universidad de Tarapacá, Arica 1000000, Chile; nmvsaez@gmail.com; 2Dirección de Gestión Digital y Transparencia, Universidad de Tarapacá, Arica 1000000, Chile; patricio.o.collao@gmail.com; 3Professional Medical School of Physical Education and Sport, Faculty of Medicine, Universidad Complutense de Madrid, 28040 Madrid, Spain; pmartinescudero@med.ucm.es (P.M.-E.); mariaelj@ucm.es (E.J.-H.); 4Electronic Engineering Department, Universidad Politécnica de Madrid, 28040 Madrid, Spain; j.pagan@upm.es; 5Center for Computational Simulation, Universidad Politécnica de Madrid, Campus de Montegancedo, 28660 Boadilla del Monte, Spain; jayala@ucm.es; 6Department of Computer Architecture and Automation, University Complutense of Madrid, 28040 Madrid, Spain

**Keywords:** oximetry, SpO_2_, artificial intelligence, machine learning, precision medicine, predictive modeling, bias assessment

## Abstract

Blood oxygen saturation (SpO_2_) is vital for patient monitoring, particularly in clinical settings. Traditional SpO_2_ estimation methods have limitations, which can be addressed by analyzing photoplethysmography (PPG) signals with artificial intelligence (AI) techniques. This systematic review, following PRISMA guidelines, analyzed 183 unique references from WOS, PubMed, and Scopus, with 26 studies meeting the inclusion criteria. The review examined AI models, key features, oximeters used, datasets, tested saturation intervals, and performance metrics while also assessing bias through the QUADAS-2 criteria. Linear regression models and deep neural networks (DNNs) emerged as the leading AI methodologies, utilizing features such as statistical metrics, signal-to-noise ratios, and intricate waveform morphology to enhance accuracy. Gaussian Process models, in particular, exhibited superior performance, achieving Mean Absolute Error (MAE) values as low as 0.57% and Root Mean Square Error (RMSE) as low as 0.69%. The bias analysis highlighted the need for better patient selection, reliable reference standards, and comprehensive SpO_2_ intervals to improve model generalizability. A persistent challenge is the reliance on non-invasive methods over the more accurate arterial blood gas analysis and the limited datasets representing diverse physiological conditions. Future research must focus on improving reference standards, test protocols, and addressing ethical considerations in clinical trials. Integrating AI with traditional physiological models can further enhance SpO_2_ estimation accuracy and robustness, offering significant advancements in patient care.

## 1. Introduction

Pulse oximetry is a widely used non-invasive method for monitoring blood oxygen saturation (SpO_2_) levels, which provides critical information about a patient’s respiratory function [1]. In various clinical settings, such as operating rooms, intensive care units, and emergency medicine, real-time monitoring of SpO_2_ is essential for timely and accurate decision-making that significantly impacts patient outcomes [2,3]. Initially, the primary method for measuring SpO_2_ was arterial blood gas analysis (BGA). Despite its accuracy, BGA is invasive, expensive, and uncomfortable for patients, limiting its practicality for continuous monitoring.

In this context, the emergence of non-invasive methods, such as pulse oximeters utilizing photoplethysmography (PPG), has revolutionized patient monitoring by providing continuous, portable, and cost-effective solutions for assessing blood oxygen saturation (SpO_2_) [4]. Pulse oximetry operates by detecting changes in blood volume within the microvascular beds, using the pulsatile nature of blood flow to measure light absorption during the systolic and diastolic phases of the cardiac cycle. By applying the ratio-of-ratios (R) technique, which calculates differential light absorption at two distinct wavelengths, PPG-based pulse oximeters provide reliable SpO_2_ readings when processed accurately [5,6].

Figure 1 shows two PPG signals captured at two different wavelengths (red and infrared) over time. The red and infrared signals vary based on the pulsatile nature of blood flow during the systolic and diastolic phases. The ratio of ratios (R), calculated as the ratio between the AC and DC components of these signals, is used to estimate SpO_2_ levels [5,6]. The exact calibration curve can vary across different pulse oximeter models, but a common general form is $SpO_{2}=110-25\cdot R$ [7]. The R value, shown in the figure as 0.50, corresponds to an SpO_2_ value of 97.5%. This calculation follows the empirical calibration curves established for the pulse oximeter [8]. Accurate interpretation of the PPG signals is essential to ensure reliable SpO_2_ estimation.

However, these non-invasive systems still face limitations, such as vulnerability to motion artifacts [4,9,10], hemoglobinopathies [11], or the need for direct skin contact, often leading to inaccuracies in low perfusion or saturation conditions or cold temperature [4,12,13,14,15]. Moreover, variations in skin pigmentation introduce bias in SpO_2_ measurements, as melanin alters light absorption, affecting the calculated R value [16,17]. Traditional oximetry systems often fail to sufficiently correct for these variations, resulting in systematic inaccuracies in individuals with darker skin tones [17,18,19,20,21]. This limitation has clinical implications due to the sensitivity of R. For instance, R=0.1 corresponds to SpO_2_ of 97.5%, while R=0.5 corresponds to SpO_2_ of 87.5%, as shown in Figure 1.

Designing accurate pulse oximeters involves several challenges. The PPG signal can be influenced by the positioning of the sensor and disturbances caused by motion artifacts, which are common and can significantly degrade signal quality [22,23,24]. Additionally, there is a lack of precise models that simulate the diffusion of light, complicating the development of expert methods for SpO_2_ measurement [25]. In critical care environments, these issues are particularly concerning, as precision and reliability in SpO_2_ measurements are vital for patient safety [26].

Despite these limitations, PPG remains indispensable in contemporary clinical practice, especially in high-risk environments where real-time, continuous monitoring is required and procedural risks must be minimized. However, interpreting PPG signals remains complex due to the influence of both physiological and non-physiological factors, necessitating the use of advanced analytical techniques to ensure measurement accuracy [27,28].

Recent years have witnessed a paradigm shift with the introduction of artificial intelligence (AI) in the analysis of PPG signals for SpO_2_ estimation [22,29,30]. AI methods, encompassing machine learning algorithms, as well as more advanced techniques such as deep learning and transformers, offer significant improvements in the accuracy and reliability of SpO_2_ measurements [31]. In clinical environments like anesthesia, intensive care, and peri-operative medicine, where continuous and real-time monitoring of oxygen levels is crucial, AI-enhanced PPG analysis could greatly enhance patient outcomes by delivering more accurate measurements [29], even under challenging conditions such as motion or poor perfusion [32]. These methods are adept at handling the inherent variability and noise in PPG signals, thus enhancing the potential for robust, non-invasive monitoring of blood oxygen levels across diverse clinical scenarios.

However, one major issue in critical care settings is the difficulty of ensuring algorithmic resilience to common clinical factors such as patient movement, varying skin tones, and diverse hemodynamic states, which affect signal quality and accuracy [11]. There is also the ongoing challenge of acquiring expansive and diverse datasets to train these algorithms, as most current datasets do not fully represent the physiological variability seen in high-risk clinical populations [33]. Moreover, generalizability across different demographic groups [19], including patients with underlying health conditions, requires further study to ensure accuracy in diverse populations in intensive care or emergency settings [34]. Additionally, existing research often lacks comprehensive evaluation metrics and offers limited validation within real-world clinical settings, underscoring the need for a more rigorous methodological approach [33,35].

Innovations in AI techniques are poised to overcome existing limitations, offering solutions that are not only more accurate but also capable of real-time monitoring in various environmental and physiological conditions [29]. The purpose of this systematic review is to explore the breadth of AI methods applied to PPG signal analysis for SpO_2_ estimation, focusing on the nuances of method types, features, preprocessing techniques, tools, libraries, and overall methodological characteristics. The review seeks to identify the current advancements in the field, highlight existing gaps, and propose directions for future research in this rapidly evolving field. Ultimately, the goal is to contribute to the development of AI-driven SpO_2_ estimation methods that are both clinically reliable and broadly accessible, with the capacity to enhance the safety and reliability of patient monitoring and improve care outcomes across a range of high-stakes clinical environments, from operating rooms to intensive care units [6,27,28].

## 2. Methodology

In this study, we performed a comprehensive review integrated with bibliometric analysis to comprehensively investigate the application of AI methods in determining blood oxygen saturation levels through PPG signals. Systematic reviews are crucial for synthesizing all relevant literature on a specific research question, providing a holistic view of the field of study [36]. Given the critical role of precise SpO_2_ monitoring in clinical settings, this review seeks to explore the current landscape of AI-driven PPG analysis within these high-stakes environments. As AI technologies evolve, their application in medical diagnostics and monitoring becomes increasingly important, especially in scenarios where timely and accurate oxygen saturation readings are crucial for patient outcomes. Coupled with bibliometric analysis, this method helps to pinpoint prevailing research trends and identify gaps in the existing literature, offering insights into the development and evolution of AI methodologies in medical research [37,38]. The primary focus of this review is to assess the effectiveness, accuracy, and reliability of various AI techniques for SpO_2_ estimation by identifying the most promising methods and highlighting areas requiring further investigation, particularly regarding how AI-based PPG analysis can be safely and effectively integrated into clinical practice.

### 2.1. Literature Search

Our systematic review covered studies up to February 2024. We adhered to the Preferred Reporting Items for Systematic Reviews and Meta-Analyses (PRISMA) guidelines to guarantee a rigorous and reproducible methodology [39]. The databases searched were PubMed, Web of Science (WOS), and Scopus, selected for their extensive coverage of medical and scientific literature.

The search strategy was developed to be both comprehensive and precise, combining keywords and Medical Subject Heading (MeSH) terms related to our research question. The query was constructed using the following terms and their variations:“oximetry” according to MeSH terms.“pulse oximetry” OR “oximet*” OR “oxygen saturation” to include all the relative references.“photoplethysmography” according to MeSH terms.“photoplethysmography” OR “PPG” as a broad term used to describe the optical imaging method for detecting arterial pulsations.“Artificial Intelligence” OR “AI” OR “Machine Learning” OR “Deep Learning” OR “Neural Networks” OR “Support Vector Machines” OR “Genetic Algorithms” OR “supervised learning” OR “unsupervised learning” to capture the broad spectrum of AI methods used in the analysis.“accuracy” OR “precision” OR “measurement Accuracy” OR “error” OR “bias” OR “reliability” to identify studies that assess the performance metrics of AI methods in SpO_2_ estimation.

The final selection process involved screening the identified records to include primary research articles that specifically addressed the use of AI methods for SpO_2_ estimation from PPG signals. Duplicates, detected by title and DOI, were removed, resulting in a total of 183 references deemed relevant for inclusion in the review. A supplemental manual search of the literature was also conducted to ensure the inclusion of all pertinent studies. The complete query is presented in Table 1.

The authors thoroughly reviewed and discussed the search algorithm to ensure the inclusion of all relevant literature. This collaborative effort aimed to minimize biases and ensure a comprehensive literature search. The query was designed to maximize the retrieval of pertinent studies while maintaining a high level of specificity to the research topic. This methodology section outlines the structured approach employed in this systematic and bibliometric review, emphasizing the rigorous process of literature search and selection based on predefined criteria and the application of the PRISMA guidelines [39].

### 2.2. Inclusion and Exclusion Criteria

We reviewed a broad range of sources, including prospective and retrospective studies, comprehensive publications, systematic reviews, conference proceedings, technical notes, and reports. The inclusion criteria focused on studies that investigated the application of AI methodologies to PPG signals for the estimation of SpO_2_. Eligible studies were required to provide detailed descriptions of the AI methods employed, feature selection, preprocessing techniques, tools, libraries, and overall methodological frameworks. In cases where studies lacked sufficient detail to meet inclusion criteria, we attempted to obtain additional information by contacting the authors directly. If no supplementary information was provided, the studies were excluded.

The exclusion criteria were as follows:Studies do not involve human subjects;Inadequately detailed AI methodologies;Literature mentions PPG signals but not in the context of estimating SpO_2_.Review articles.

### 2.3. Data Extraction and Analysis

A well-crafted query was designed to optimize results from Scopus, PubMed, and WOS. Two authors (NS and AMC) independently conducted the quality and eligibility assessments of relevant references. Additionally, the bibliographies of the identified references were reviewed to include related studies. Titles, abstracts, and author keywords were initially analyzed independently to assess eligibility, followed by a thorough examination of the full texts to confirm they met the inclusion criteria.

To facilitate visual representation and bibliometric analysis, the word cloud included in the abstract and the summary metrics presented in Table 2 were generated using the Bibliometrix package, an R-based software designed for comprehensive bibliometric studies [40]. The study quality assessment, described in Section 3.6, was conducted using in-depth evaluation methods, while the robvis web application was used specifically for visualizing the results of the risk-of-bias assessments [41].

Figure 2 illustrates the multi-stage eligibility screening conducted during the search process. Articles were excluded if they focused on predicting or characterizing specific diseases rather than SpO_2_ estimation, even if they discussed using SpO_2_ and AI methods. Out of the 187 records screened (abstracts and titles), we excluded 141 because these records did not focus on AI methodologies specifically applied to PPG signals for SpO_2_ estimation. After titles and abstract screening, 46 references were identified for further detailed examination in the full-text stage. References that failed to meet the inclusion criteria were subsequently excluded. The primary reasons for exclusion were inadequately described AI methods (4); studies not focused on SpO_2_ estimation from PPG signals (10); clinical validation of prototypes (6); and review papers (2). We also added 2 articles through snowballing [42,43].

### 2.4. Data Collection and Outcomes

Data were meticulously gathered by NS and AMC through a thorough review of each article’s full text. This included parameters such as AI models, methods and tools employed, libraries, extracted features, preprocessing techniques for PPG analysis, performance metrics, number of participants, clinical/normal study classification, gender, age, skin color classification method or demographic information, type of oximeter employed, databases utilized, range of SpO_2_ saturation intervals tested, main findings, and limitations.

The main objective of this study involved assessing the performance of various SpO_2_ estimation methods based on a critical review of included/excluded parameters in each study and analyzing the stated performance indicators, their real-world relevance, and limitations. A secondary outcome was to showcase the range of available SpO_2_ estimation methods, including face-based non-contact techniques employing various imaging modalities, and to compare them with corresponding reference devices. Due to the heterogeneity among SpO_2_ estimation studies, different performance metrics were independently chosen for analysis, with Mean Absolute Error (MAE) and Root Mean Square Error (RMSE) being the most frequently reported metrics.

This review highlights significant trends, gaps, and challenges in the current research landscape of AI-driven SpO_2_ estimation from PPG signals, offering insights into future directions for innovation in non-invasive blood oxygen monitoring techniques. A notable challenge highlighted across studies is the lack of a consistent ground truth for oximetry, as most studies opted for less invasive methods over the more accurate yet invasive blood gas analysis.

## 3. Results and Discussions

### 3.1. Characteristics of the Selected Studies

Table 2 summarizes key metrics from the selected articles on AI-based SpO_2_ estimation, spanning from 2008 to 2024. It includes 26 documents from 22 sources, with an average of 11.64 citations per document and 4.14 citations per year per document, reflecting their academic impact. The studies reference 797 works, with contributions from 131 authors and no single-authored documents, highlighting the collaborative nature of the research. Additionally, the analysis covers 391 unique keywords, emphasizing the breadth of topics explored in the field.

Table 3 encapsulates the progression and diversity of research in AI-based SpO_2_ estimation, from traditional devices to innovative non-contact methods. Each entry provides the first author’s name, publication year, citation, and database or type of oximeter used in the study, and the range of SpO_2_ measured when specified.

The studies span from 2008 to 2024, demonstrating the evolution of technologies and methodologies used for SpO_2_ estimation. Traditional pulse oximeters, such as the Masimo Radical and Nellcor models, are frequently employed alongside modern wearable technologies and smartphone-based systems. Various brands, including CMS-50E, EMAY EMO-80, MAX30102, and unconventional devices like the Samsung Galaxy S21 Ultra, underscore the diverse array of technologies utilized in SpO_2_ estimation research. Notably, the Masimo Radical oximeter is recognized as a benchmark in some studies, although its precision might fall short of clinical standards.

These studies explore a variety of AI techniques, demonstrating the potential for these methods to reduce the impact of confounding factors, such as motion artifacts and variations in perfusion. These advances could significantly improve patient safety by providing more accurate and reliable oxygen saturation measurements, reducing false alarms, and supporting clinical decision-making in time-sensitive environments.

A significant observation across the studies is the absence of a consistent ground truth for oximetry measurements. Most studies opt for less invasive methods, avoiding the gold standard of BGA. Notably, only three studies [43,49,50] included BGA, posing a challenge in validating AI-derived oximetry data against a true medical standard.

Several studies utilize large, publicly available datasets, such as the BIDMC PPG and Respiration Dataset [56,60] or the MIMIC II matched waveform database [46,61]. These datasets serve as invaluable repositories for respiratory and PPG signal investigations. The inclusion of datasets like the PURE Dataset [54] and UBFC [63] exemplifies efforts to access varied physiological data, enriching the scope and applicability of AI methodologies for SpO_2_ estimation.

Regarding the use of wrist-worn devices, these devices are preferred by users for their convenience, despite the wrist being a suboptimal measurement site compared to the finger or earlobe, which have higher capillarity [10]. In clinical settings, accurate non-invasive SpO_2_ monitoring is critical, yet consumer-grade devices often fall short in precision compared to medical-grade pulse oximeters [64]. This discrepancy limits their use in critical care environments, such as anesthesia and intensive care units, where precise real-time monitoring is essential.

Reflectance-based oximeters, as highlighted in several studies [22,23,43,49,50,59], and fingertip devices underscore their significance in non-invasive SpO_2_ monitoring. Some studies utilized non-contact methods of measuring oxygen saturation, specifically mentioning rPPG (remote photoplethysmography) [25,54,55,60,63]. The inclusion of rPPG (remote Photoplethysmography) technologies represents a significant advancement in oximetry, enabling more user-friendly, non-invasive monitoring solutions that can be integrated into everyday devices like smartphones and smartwatches [65]. These non-contact methods for estimating HR and SpO_2_ are particularly advantageous in continuous monitoring scenarios or environments where traditional contact-based sensors might be impractical or uncomfortable [54]. This advantage is especially noted in studies using smartphone cameras [42,62], wearable smartbands [47,59], or other wearable multi-channel PPG prototypes [51,52,54].

These methods rely on video acquisition using cameras or other optical sensors to capture regions of interest (ROIs) on the face or other body parts without direct skin contact [66]. However, the clinical accuracy of rPPG-based SpO_2_ estimation remains under investigation, as various factors such as camera type, video resolution, frame rate, and shooting distance impact the reliability of these methods [67]. The type of camera, defined by the color channels used for video acquisition, plays a crucial role in determining the quantity and quality of PPG information available for physiological parameter estimation. RGB spectra, for example, demonstrate stronger pulsatile strength compared to infrared spectra, making them more effective for extracting accurate PPG signals in clinical monitoring applications [67].

In rPPG studies, the ROIs are multiple regions on the face, excluding the eye area or mouth to avoid interference from blinking or lip movement [25,54,55]. This careful selection is particularly important in peri-operative and intensive care settings, where facial obstructions such as masks or medical devices may pose challenges to reliable monitoring. To mitigate issues such as asymmetric lighting, head rotation, or skin occlusion due to facial hair or masks, these studies focus on selecting the most effective ROI for vital sign calculation [54]. For example, Zhu et al. segmented the face using the Simple Linear Iterative Clustering (SLIC) approach to ensure uniform light reflection characteristics [55]. Kim et al. converted the image to the YCbCr color space for skin pixel clustering [25], and Shuzan et al. dynamically tracked and segmented multiple facial regions according to spatial and color proximity [60].

Recently, Peng et al. further advanced this field by proposing an SpO_2_ analysis architecture that combines unsupervised contrastive learning with supervised label learning to obtain accurate rPPG signals from remote videos [63]. Their approach focused on extracting ROIs from hands and specific facial regions, such as the forehead, cheeks, and nose, in RGB videos detected using the YOLO algorithm [68], which effectively restored rPPG signals with enhanced interpretability. Additionally, they developed an attention module based on spatiotemporal manual features, significantly improving the efficacy of rPPG feature utilization for blood oxygen estimation. This innovation underscores the potential of integrating advanced AI techniques with rPPG technology, especially in clinical environments like intensive care or peri-operative settings, to enhance the accuracy and reliability of non-invasive SpO_2_ monitoring.

It is worth remarking that assessing AI models for SpO_2_ estimation from PPG signals requires testing across a broad spectrum of saturation intervals, from normal oxygen levels to hypoxia, to evaluate their performance in critical clinical scenarios. This diversity is crucial for determining model accuracy and robustness in various medical situations, where maintaining oxygen saturation levels is vital. Simulating a comprehensive range of SpO_2_ levels, especially under hypoxic conditions, presents ethical challenges [13]. To address these challenges in clinical research, techniques such as administering supplemental oxygen, using mechanical ventilation, or inducing physical exertion are employed [69]. However, the latter may introduce motion artifacts that compromise PPG signal integrity [70].

The saturation intervals reported in studies vary widely, typically ranging from 70 to 100% [23,43], with some studies reporting narrower ranges such as 93–100% [63] or 90–98% [25]. Another study covers broader ranges, such as 60–100% [50]. Accurate monitoring within these intervals is crucial for ensuring patient safety in scenarios where SpO_2_ fluctuations can indicate life-threatening complications.

Conducting accurate model evaluations requires encompassing a full spectrum of SpO_2_ values, including hypoxic ranges. However, replicating such conditions naturally in human subjects is ethically and medically challenging. Therefore, researchers must balance the need for comprehensive testing with the ethical implications of inducing hypoxia in study subjects, while also ensuring that models are robust enough to be used in real-world clinical environments, such as emergency medicine or peri-operative care.

### 3.2. Estimation Methods

In our systematic review, we categorized the AI models used for SpO_2_ estimation from PPG signals based on a framework derived from the scikit-learn taxonomy of supervised learning models [71]. This approach allowed us to classify the diverse models into distinct categories, including Linear Models (LMs), Neural Network Models (NNMs), Ensemble Models (EMs), and other methods such as Support Vector Machines (SVMs), Decision Trees (DTs), and K-Nearest Neighbors (KNN). Each category reflects different strengths and applications, ranging from simpler linear models to advanced neural networks like CNNs, which are better suited for capturing complex patterns in physiological data. Figure 3 illustrates this classification scheme, providing a structured overview of the AI models employed in clinical monitoring and experimental settings while highlighting trends and preferences in pulse oximetry research.

Linear Models (LMs) are the most frequently utilized, reflecting their simplicity and effectiveness in handling linear relationships within the data. Neural Network Models (NNMs) and Ensemble Models (EMs) also feature prominently, indicating their robustness in capturing complex patterns and enhancing prediction accuracy. Specifically, our analysis encompasses various models, such as:**Neural Network Models (NNMs)**: Neural networks, particularly Artificial Neural Networks (ANNs) [48,56] and Convolutional Neural Networks (CNNs) [23,42,57,58,62], are among the most widely used models for SpO_2_ estimation. These networks excel at identifying complex patterns in physiological data, making them highly suited for clinical monitoring tasks. Variants like 3D CNNs [31], U-Net [61], and CL-SPO2Net [63] are tailored for specific datasets or device inputs (e.g., smartphone sensors like those found in iPhone models) [42]. Neural networks, particularly CNNs, have shown great potential in analyzing both raw PPG signals and video-based data for real-time SpO_2_ estimation, a crucial aspect in high-stakes environments such as emergency medicine, operating rooms, and ICUs.**Support Vector Machines (SVMs)**: Diverse SVM applications include Support Vector Machines Regression for traditional regression tasks [44,60] and specialized implementations such as SVMs for classification across subjects [46] or SVMs with linear kernels [22], highlighting their versatility in handling both linear and non-linear data in high-stakes clinical settings.**K-Nearest Neighbors (KNN)**: Used for both classification and regression [31,43,55,60], relying on proximity-based predictions, which can be valuable for medical data with inherent patterns related to patient groups.**Decision Trees (DTs)**: Enhanced by methods like Bagged Trees [60] and Ensemble Trees [43,60], which aggregate predictions to improve accuracy, these models are often favored in clinical settings for their interpretability and ease of use.**Ensemble Methods (EMs)**: Techniques like Random Forests [43,47,50], Gradient Boosting [43,47,58], and AdaBoost [58] leverage the power of combining multiple models to enhance prediction accuracy and stability. This is crucial for medical applications, where reliability and robustness are paramount. More sophisticated ensemble approaches like XGBoost [47] and Extra Trees Regressor [58] reflect the need to optimize performance for critical clinical decision-making.**Gaussian Processes (GPs)**: Gaussian Process regression models have demonstrated strong performance in estimating respiratory rate (RR) and oxygen saturation from PPG signals, outperforming other machine learning models for both RR and SpO_2_ estimation [60]. This probabilistic approach provides a measure of uncertainty in predictions, crucial for clinical applications thanks to their ability to handle small datasets and provide uncertainty estimates.**Explainable AI (XAI)**: Although not an AI model itself, XAI is a framework or set of techniques aimed at making AI model outputs more understandable to humans. XAI is included in our review due to its importance in medical AI systems, where explainability is critical. Clinicians need to understand how decisions are made, and XAI techniques help interpret model outputs by providing insights into the contribution of each feature to the prediction. This enhances transparency and builds trust in AI-based clinical monitoring systems [59].**Transformer Models, (TFM)**: Transformer models have shown great potential across various clinical applications due to their ability to handle sequential data efficiently, making them highly suitable for continuous monitoring in intensive care units (ICUs) and operating rooms, where real-time data analysis is crucial [61]. They are increasingly being used in tasks such as patient monitoring, medical imaging, and even predicting clinical outcomes, thanks to their powerful ability to model long-term dependencies and process large datasets with high accuracy. An example of this is the Vision Transformer (ViViT) architecture, which has been adapted for video-based SpO_2_ estimation, demonstrating the versatility of transformers in handling visual and time-series data for physiological monitoring applications [58]. Transformers are particularly valuable for their scalability and ability to integrate multimodal data, making them a versatile tool in modern healthcare settings.

This distribution reflects a strong interest in employing a wide range of AI techniques to maximize the information potential of PPG signals in both clinical and non-clinical environments. The diverse methodologies for SpO_2_ estimation illustrate the field’s complexity and the ongoing innovation aimed at enhancing the accuracy and reliability of non-invasive monitoring of blood oxygen levels.

Figure 4 presents the distribution of models across three main categories: Linear Models (LMs), Neural Network Models (NNMs), and Ensemble Models (EMs), offering insight into the diverse approaches used for SpO_2_ estimation and medical data analysis. The figure highlights the variety and prevalence of different AI models within each category.

In the Linear Model (LM) category, Lasso and Linear Regression are the most commonly used, accounting for over 30% of the approaches. These models are favored for their ease of implementation and interpretability, particularly when dealing with straightforward linear relationships in PPG data [31,43,58,60]. However, the growing complexity of medical data has led to the use of more advanced linear models like Ridge Regression, Quantile Regressor, and ElasticNet, which aim to capture more intricate patterns in non-linear data. Despite their simplicity, linear models face limitations in critical care settings where complex physiological signals require more sophisticated methods for accurate interpretation.

Neural Network Models (NNMs) show a significant preference for Convolutional Neural Networks (CNNs), which make up 80% of the models used. Well-known CNN architectures such as U-net [61], ResNet [54], and VGG-19 [22] are particularly effective in tasks like image segmentation and classification, which are crucial for interpreting physiological data from PPG signals. These CNNs are widely applied in both clinical environments (e.g., ICUs) and controlled experimental settings, involving healthy participants like college students and teenagers [22,42,43]. Artificial Neural Networks (ANNs) and Recurrent Neural Networks (RNNs), while less frequently used (accounting for 20%), are employed in specialized tasks involving time-series data, such as real-time SpO_2_ estimation and signal quality assessment. The dominance of CNNs in this category highlights their importance in a variety of settings, from clinical to non-clinical applications.

For Ensemble Models (EMs), Random Forests are the most utilized, comprising 35.7% of the ensemble approaches. These models are particularly valued for their ability to combine the outputs of multiple decision trees to improve predictive performance [31,43,47,50]. Gradient Boosting methods, such as XGBoost, and Bagging Methods, like bootstrap aggregation, are also commonly used due to their efficiency and scalability, especially in high-risk clinical settings such as studies involving mechanically ventilated children or patients with pulmonary disorders [48,50]. Ensemble models have also found success in non-clinical environments, such as studies involving teenagers, to assess physiological data [47]. The robustness and reliability of ensemble models make them especially valuable in clinical environments where accuracy and stability in decision-making are crucial [58].

### 3.3. Key Performance Metrics

Comparing the performance of different models is essential for identifying the approaches that yield the most accurate and reliable SpO_2_ estimations. Mean Absolute Error (MAE) and Root Mean Square Error (RMSE) are the primary metrics used to assess prediction errors. Other important metrics include Mean Absolute Percentage Error (MAPE), standard deviation (SD), percentage accuracy, F-score, and the coefficient of determination ($R^{2}$). These provide a comprehensive evaluation of the models’ performance, while additional factors like the number of features, sensitivity or recall, median correlation, best epochs, and losses (train, test, and validation) offer insights into the complexity and optimization of SpO_2_ estimation models based on PPG signals.

Focusing on the most frequently reported metrics ensures relevant comparisons. In this review, we emphasize the most commonly used metrics: MAE and RMSE. As per international standards, SpO_2_ estimation should achieve a Mean Absolute Error (MAE) of no more than 2% and a Root Mean square Error (RMSE) of up to 3.5%, particularly within the SpO_2_ range of 70% to 100%. Reaching this level of precision generally necessitates clinical trials where healthy subjects are deliberately brought to low oxygen saturation levels, a process that is not representative of real-world conditions and introduces ethical challenges [33,72]. Moreover, the limited availability of labeled data further complicates the validation process. This underscores the importance of AI techniques in analyzing PPG signals, which offer promising solutions to overcome the limitations of traditional methods.

Figure 5 presents the distribution of Mean Absolute Error (MAE) percentages across various AI model categories using boxplots. GPs and EMs emerge as the top performers, with GPs showing a median MAE close to 0.95% and very low variability, indicating consistent and reliable performance. EMs also demonstrate strong performance with a median MAE around 1.04%, though with slightly higher variability compared to GP.

LMs and NNMs provide moderate results. LMs have a median MAE of about 1.36%, reflecting reasonable accuracy with some variability. NNMs, while powerful, shows a higher median MAE of around 1.97% and greater variability, highlighting the need for careful tuning to achieve optimal results.

SVMs exhibit considerable variability with a median MAE of approximately 1.9%, indicating that their performance is highly dependent on the specific application and dataset. DTs and KNN also show variable performance. DTs have a median MAE around 1.33% but with a significant spread, suggesting sensitivity to data. KNN, although represented by fewer data points, shows promise with a median MAE of about 1.04%.

The TFM and XAI categories, each represented by limited data, indicate good performance with MAE values around 0.75% and 1.12%, respectively. However, the limited sample size warrants a cautious interpretation of their effectiveness.

Regarding outliers, several categories exhibit notable outliers, particularly within the EM, SVM, and DT groups. These outliers indicate instances where the models performed significantly worse than their typical range, highlighting the importance of considering these variations in performance when selecting a model. The presence of outliers in the NNM and LM categories further underscores the variability and potential for inconsistent results within these model types.

Figure 6 illustrates the distribution of Root Mean Square Error (RMSE) percentages across various AI model categories used for SpO_2_ estimation from PPG signals. The figure provides valuable insights into the performance and variability of different model categories.

EMs and GPs demonstrate the best performance, characterized by low and consistent RMSE values. EMs have a median RMSE close to 1.40%, though with some variability, while GP models show minimal variability with an average RMSE of around 1.45%, indicating high reliability and accuracy.

LMs and DTs also perform well, with median RMSE values around 1.7% and 1.8% respectively. These models show slightly higher variability compared to GPs but still maintain a reasonable level of consistency in their predictions.

NNMs, although powerful, exhibit higher median RMSE values around 2.7% with significant variability, suggesting that while some neural networks may perform well, others might not generalize as effectively, leading to less consistent results. SVM and KNN show moderate performance with median RMSE values around 1.8% and 1.5%, respectively, but also with notable variability.

Several categories exhibit notable outliers, particularly within the EM, SVM, and DT groups. These outliers indicate instances where the models performed significantly worse than their typical range, highlighting the importance of considering performance variations when selecting a model. The presence of outliers in the NNM and LM categories further underscores the variability and potential for inconsistent results within these model types.

Additionally, the number of models tested in each category significantly affects performance variability and error margins. Categories with a greater number of models, such as Linear Models and Ensemble Models, tend to exhibit higher variability and larger error margins, contributing to the presence of outliers. This variability may be attributed to potential overfitting or the small sample sizes (N) used in some studies. For example, studies by Aguirregomezcorta and Kim reported sample sizes of only 10 participants [25,43], while Mathew had 14 participants [57], potentially leading to less reliable performance estimates. In contrast, larger studies, such as those by Chu (1732 participants) [61] and Guo (513 participants) [47], are more likely to provide robust results with reduced variability. Smaller sample sizes can result in less reliable performance estimates and increased susceptibility to overfitting. Therefore, it is crucial to consider the sample sizes reported in the studies to provide context for these findings and ensure that the observed performance metrics are robust and generalizable.

In machine learning, model performance is typically evaluated using two key data subsets: the validation set and the test set. The validation set is a portion of the data used during the training process to fine-tune the model’s parameters and help prevent overfitting (where the model learns too closely from the training data and performs poorly on new data). The test set, on the other hand, is a completely separate portion of the data that is used only after the training process is complete. It simulates how the model will perform on entirely new, unseen data in real-world applications. Therefore, discrepancies in performance between the validation set and the test set can indicate that a model is not generalizing well beyond the data it was trained on.

For instance, the Random Forest Regressor showed significantly worse performance on the test set (MAE = 6.56%, RMSE = 8.21%) compared to the validation set (MAE = 1.25%, RMSE = 1.25%) [50]. Similarly, a CNN showed higher errors on the test set (MAE = 3.28%, RMSE = 3.69%) compared to the validation set (MAE = 0.75%, RMSE = 0.99%) [57].

These discrepancies suggest that the models are overfitting to the training data and do not generalize well to new, unseen data. This poor generalization may be attributed to inherent physiological differences among patients, which are not adequately represented in the training set [50]. Additionally, small dataset sizes and lack of standardized protocols can result in a narrow dynamic range of SpO_2_ values, further affecting model performance [57]. Validating models against gold standards, such as arterial blood gas analysis (BGA) or well-known FDA-approved devices, is essential. Using another suboptimal pulse oximeter as the reference can limit the potential accuracy of AI models.

These factors contribute to models that perform well on validation data but struggle to maintain their performance on test data. This highlights the importance of developing more robust training datasets, refining feature selection, as will be discussed in the next subsection, and validating against accurate standards to enhance model generalization and effectiveness in clinical applications.

### 3.4. Feature Extraction

Feature extraction simplifies raw physiological data by isolating the most relevant information, enhancing model efficiency and performance. Selecting the right features improves AI model accuracy and ensures effective generalization for real-time patient monitoring. In SpO_2_ estimation, these features offer valuable insights into underlying physiological processes, crucial for critical care.

Preprocessing techniques play a crucial role in improving the quality of the PPG signal before feature extraction, particularly in high-risk environments such as operating rooms and ICUs. These techniques address common challenges like noise and motion artifacts, which are prevalent in clinical settings. Signal quality assessment (SQA) is vital to ensure the input data for AI models is clean and reliable, directly impacting the accuracy of SpO_2_ estimations. Poor input data quality can lead to inaccurate feature extraction and unreliable model predictions, potentially jeopardizing patient safety. Ensuring high-quality PPG signals through rigorous SQA is thus fundamental to developing robust AI models for SpO_2_ estimation, particularly in time-sensitive clinical applications [73,74].

Filtering techniques such as Butterworth filters are commonly used to isolate relevant frequency components and remove noise from the signal, which is especially important when dealing with electronic interference in medical equipment [75]. For instance, Badiola-Aguirregomezcorta et al. used a low-pass filter (10 Hz), a fourth-order band-pass filter (0.67–4.5 Hz), and a sixth-order low-pass filter to separate AC and DC components of the PPG signal and ensure the PPG signals were clean and reliable for SpO_2_ estimation [43]. Guo et al. employed a band-pass Butterworth filter (0.5–5 Hz) [47], while Kim et al. used a zero-phase fifth-order Butterworth band-pass filter [52]. Ding et al. applied a band-pass filter (0.7–4 Hz) combined with a Savitzky–Golay filter for low-pass filtering [42]. Venkat et al. introduced a wavelet-based detrending method to mitigate baseline wander in PPG signals [49].

Statistical features such as standard deviation, variance, skewness, and kurtosis are frequently used are and reported in five studies [22,23,47,56,58]. These measures provide fundamental insights into the variability and central tendency of PPG signals, which are critical for assessing signal quality in clinical monitoring.

Temporal features, derived from the time-domain representation of the PPG signal, include peak-to-peak intervals, pulse duration, and pulse amplitude. Frequency domain features, extracted through techniques like the Fourier Transform, include power spectral density and dominant frequencies. Shuzan et al. analyzed features across the frequency, time, and spatial domains, highlighting the multidimensional nature of signal processing in SpO_2_ estimation [60].

Morphological features, which examine the waveform shape and structure (e.g., the contour of PPG waveforms and the slope of signal segments), were explored in two studies [44,45], demonstrating their value in capturing the dynamic characteristics of blood flow in critical care. Additionally, analyzing the AC and DC components of PPG signals, which reflect pulsatile and non-pulsatile blood flow, is a method highlighted in two studies [43,49]. This decomposition of signals into interpretable parts is essential for estimating SpO_2_ levels accurately in patients.

Ratios and the ratio of ratios (RoR), which capture relative changes in PPG signal components, were emphasized in three studies [51,55,60], underscoring their effectiveness in linking PPG signals to SpO_2_ levels in complex clinical scenarios.

Advanced methodologies such as Principal Component Analysis (PCA) have been employed for dimensionality reduction in two studies [46,63]. PCA simplifies data complexity by transforming it into orthogonal features, allowing for more efficient data processing. Additionally, libraries such as tsfresh have been used to automate the extraction of relevant features from time-series data, streamlining the feature extraction process [59]. These advanced techniques play a pivotal role in simplifying complex datasets and revealing patterns critical to accurate SpO_2_ estimation in clinical settings.

In one notable study, Shuzan et al. employed multiple machine learning models and nine feature selection algorithms after extracting 107 features from the PPG waveform [60]. The choice of model and feature selection algorithm significantly impacted the results. For example, the GP model with the top 11 features selected by the ReliefF algorithm achieved an RMSE of 0.98% and an MAE of 0.57%, demonstrating high accuracy. In contrast, the SVM model using the top 28 features selected by the CFS algorithm resulted in an RMSE of 5.66% and an MAE of 2.19%, indicating lower performance, while another model in the same study showed an RMSE of 5.58% and an MAE of 2.04%. This variation underscores the importance of appropriate feature selection in achieving optimal model performance for SpO_2_ estimation.

Mathew et al. conducted two ablation studies to justify using nonlinear channel combinations and convolutional layers for temporal feature extraction. The first study compared nonlinear to linear channel combinations, showing that the nonlinear approach achieved better performance. The second study compared convolutional layers to fully connected layers, demonstrating that convolutional layers provided superior results for temporal feature extraction [57]. The same finding was reported by Gammariello et al., stating that nonlinear channel combinations and convolutional layers for temporal feature extraction improve the performance of SpO_2_ estimation models [62].

Explainable AI models, such as SHAP (SHapley Additive exPlanations) values, have played a significant role in improving the transparency and trustworthiness of SpO_2_ estimation models. In the study by Zhong et al., SHAP was applied to explain subject clustering and SpO_2_ regression models [59]. The researchers first trained a supervised binary classifier to differentiate between clusters using 248 features, then calculated SHAP values for each feature. This approach provided clear insights into feature contributions, enhancing model interpretability and fostering greater trust in AI-driven clinical decisions.

For instance, SHAP values were instrumental in identifying key features such as partial auto-correlation, autoregressive coefficients [76], approximate entropy [77], and kurtosis of FFT magnitudes [51]. Notably, higher values of “kurtosis of FFT magnitudes” were linked to higher SpO_2_ levels, indicating that dominant frequency bands with energy spikes correspond to better oxygen saturation [59]. This level of explainability is crucial in clinical environments, where understanding the rationale behind model predictions can support more informed decision-making by healthcare professionals.

The use of explainable AI models in critical care settings allows for both the optimization of patient monitoring systems and the validation of model predictions against well-established clinical standards. SHAP values and similar interpretability techniques will continue to play a critical role in making AI models more transparent, offering insights into the relationships between PPG signal features and SpO_2_ values.

The diversity in feature selection and extraction algorithms highlights the complexity of PPG signal interpretation and the innovative strategies researchers are developing to address these challenges. These approaches not only streamline complex data but also reveal critical patterns for SpO_2_ estimation, laying the groundwork for further scientific exploration and clinical innovation.

### 3.5. Software Libraries and Tools

In our review, we found a diverse range of software libraries and tools used across various studies. These tools were crucial for tasks such as signal processing, machine learning, and feature selection. For example, the Python toolkit HeartPy was frequently used for cardiac signal processing in studies like the one by Koteska et al., which employed a Deep Artificial Neural Network to estimate SpO_2_ from PPG signals [56]. Liu et al. also utilized Python libraries for implementing SVM and CNN models to evaluate the quality of PPG signals on wearable devices [22].

Machine learning tasks often relied on popular libraries such as scikit-learn, which was used by Ghazal et al. for balancing data using techniques like TOMEK links and SMOTE [48]. TOMEK links remove noisy data points from the majority class that are close to the decision boundary, helping to refine the separation between classes. SMOTE (Synthetic Minority Over-sampling Technique) generates synthetic examples of the minority class to balance the dataset, improving model performance on imbalanced data [78].

Deep learning models were predominantly implemented using TensorFlow and PyTorch [42,54]. For instance, Qiao et al. and Chu et al. leveraged these frameworks for deep learning models aimed at vital sign measurement and SpO_2_ estimation, respectively [54,61].

MATLAB’s Feature Ranking Library (FSLib) and Hyperopt were commonly used for these feature selection and model optimization. Shuzan et al. applied MATLAB for feature ranking in their machine learning-based respiration rate and SpO_2_ estimation study [60], while Guo et al. used Hyperopt for hyperparameter optimization in an XGBoost-based physical fitness evaluation model [47]. These tools allowed for efficient model tuning and feature selection, enhancing the models’ predictive accuracy.

Another significant development in this field is the integration of cloud-based analytics and real-time notifications to enhance biomedical monitoring systems. Gayatri et al. designed a DNN-based system that interfaced sensors with a Raspberry Pi-4 board, achieving higher accuracy using TensorFlow and Keras [53]. This system supported continuous monitoring of heart rate and SpO_2_ parameters, with data stored on Thinkspeak for cloud analysis and real-time notifications sent via Twilio [79]. The integration of these platforms underscores the growing importance of real-time data analytics and remote monitoring in clinical care.

The frequent citation of GitHub links and the use of open-source libraries highlight a commitment to transparency and replicability in research. Vijayarangan et al. provided their code on GitHub for robust modeling of reflectance pulse oximetry, further demonstrating the trend toward open science in biomedical research [50]. This practice enhances collaboration and enables other researchers to replicate and build upon existing studies.

Generalized Additive Models (GAMs), which allow for the modeling of non-linear relationships between a response variable and predictor variables without assuming a specific distribution, have proven to be a powerful tool for analyzing complex data, such as PPG signals [58].

Overall, the diverse use of software libraries and tools across studies reflects the innovative approaches researchers are taking to improve the accuracy and reliability of SpO_2_ estimation from PPG signals. This diversity also highlights the collaborative nature of the field, where different methodologies and tools are combined to tackle complex biomedical challenges effectively.

### 3.6. Study Quality Assessment

Finally, we performed a meticulous quality assessment of the included studies using a tailored scoring scheme designed for this review. This assessment focused on key parameters and performance metrics relevant to AI-driven SpO_2_ estimation from PPG signals. Although there is no standard risk of bias tool for methods-comparison studies, we followed the QUADAS-2 protocol, which evaluates four domains: patient selection, index test, reference standard, and flow and timing of patients [80].

First, one author (NS or AMC) independently assessed the risk of bias for the selected references. Then, another author (POC, JP, or PE) reviewed the assessment. Discrepancies were resolved through discussion. Finally, we used the robvis tool to visualize the risk of bias assessment [41].

Figure 7 illustrates the risk of bias assessment for each of the included studies using the QUADAS-2 protocol, adapted for evaluating AI-driven SpO_2_ estimation from PPG signals. The color coding used in the figure is as follows: Red indicates a high risk of bias, yellow indicates some concerns, and green represents a low risk of bias. The figure categorizes the risk of bias into four domains: patient selection, index test, reference standard, and flow and timing, along with an overall judgment.

**D1: Patient selection**. This domain assessed whether studies provided sufficient demographic information (age, gender, number of participants, skin color, or ancestry reported) and type of participants (clinical/controlled environment). For instance, Kim et al. had a small sample size of 10 healthy participants [25]. In contrast, larger studies such as Guo et al. (reported 513 participants) or Chu et al. (included 1732 participants) reported the largest datasets, providing highly robust data for clinical applications [61].**D2: Index test (AI model and PPG analysis).** This domain evaluated the methods or tools employed, preprocessing techniques for PPG analysis, and extracted features. In several studies, the use of filters and preprocessing tools such as Butterworth filters was highlighted. For instance, Shuzan et al. extracted features from a filtered PPG signal for accurate SpO_2_ estimation [60]. The AI models used across the studies varied, with some relying on traditional machine learning techniques, such as Support Vector Machines (SVMs) in the study by Liu et al. [22], while others employed advanced deep learning models.**D3: Reference standard.** This domain analyzed whether the studies employed a ground truth method for SpO_2_ estimation and considered sample size. Some studies used traditional pulse oximeters as the reference standard. However, others, such as Aguirregomezcorta et al., used arterial blood gas analysis (BGA), a more accurate, gold-standard method for reference [43]. The variety of reference standards emphasizes the importance of validating AI models against reliable, consistent methods to ensure clinical accuracy.**D4: Flow and timing.** This domain evaluated the range of SpO_2_ saturation intervals tested and the reporting of performance metrics such as MAE and RMSE. Chu et al. included a large range of saturation intervals with over 1732 participants, providing key performance metrics and robust data for SpO_2_ estimation [61]. By contrast, Vijayarangan et al. tested reflectance pulse oximetry with a narrower saturation interval and a smaller sample size [50].

The scoring scheme was developed to be both comprehensive and intuitive, following a two-step process. First, we assigned scores based on the inclusion or exclusion of factors such as methods, tools, libraries, extracted features, preprocessing techniques for PPG analysis, performance metrics, demographic information, type of oximeter employed, databases utilized, range of SpO_2_ saturation intervals tested, clinical/normal study classification, and skin color classification methods. Each category was classified based on the completeness of the reported data. Categories were considered ”low” if all relevant data were reported, “some concerns” if some data were missing, and “high” if significant data were absent.

Second, we estimated the overall classification by averaging the scores across the four domains. The overall classifications were defined as follows: “low” for a score of 2 points or less, “some concerns” for 3 to 4 points, and “high” for scores of 5 points or more. This structured approach to quality assessment ensures a systematic evaluation of the methodological robustness and reporting comprehensiveness of each study. By carefully analyzing these domains, we provide a clearer understanding of the current state of AI applications in non-invasive SpO_2_ estimation from PPG signals, highlighting areas that require improvement, particularly in the context of robust validation and the inclusion of more diverse patient populations.

The analysis reveals that only 3 out of the 26 studies (11.5%) have a low risk of bias across all domains [43,50,60,61]. Conversely, 8 out of the 26 studies (30.8%) exhibit a high risk of bias in at least two domains [22,25,42,47,51,53,54,58], while some concerns were noted in 10 studies (38.5%).

In the patient selection domain, D1, a high risk of bias was observed in 6 studies (23.1%) [23,44,45,51,53,54], some concerns were identified in 17 studies (65.4%), and a low risk of bias was found in 3 studies (11.5%) [57,60,61]. These three studies reported all demographic information, also including a method for skin color classification, which is crucial for the accuracy measurement of the SpO_2_ [18,81].

When assessing the bias related to patient selection, the inclusion of both clinical and controlled environments is crucial. Studies involving non-healthy patients (23.1%), such as critically ill children [48], pulmonary patients [49], and critically ill adults in ICU settings from the BIDMC and MIMIC databases, generally demonstrated better methodological rigor. The BIDMC database, for instance, includes over 10,000 ICU patient records [56,60], while the MIMIC database comprises over 60,000 ICU admissions, offering substantial data to support AI model development and validation [46,61].

In the index test domain, D2, all studies demonstrated a low risk of bias, indicating that AI models and PPG analysis techniques were generally well-described, validated, and appropriate across the reviewed literature.

For the reference standard domain, D3, only three studies [43,49,50] included the arterial blood gas analysis method as the ground truth for SpO_2_ estimation, which is considered the gold standard. These studies, with sample sizes of 95, 28, and 10 participants, respectively, provided robust validation, thus reducing the risk of bias in this domain. Despite the small sample size of 10 subjects in Badiola-Aguirregomezcorta et al. [43], which might initially be considered at high risk of bias due to the limited number of participants affecting the generalizability of the findings, the use of the BGA method and the challenges associated with conducting such studies support its classification as having a low risk of bias.

Studies involving non-healthy patients generally reported lower risks of bias due to their large sample sizes and robust methodologies. However, studies with smaller sample sizes, like Ogawa et al. (three subjects) [44], presented a higher risk of bias due to the potential lack of representativeness and validation rigor. Overall, a high risk of bias was observed in 15 studies (57.7%), emphasizing the widespread challenge of using inadequate ground truth methods for SpO_2_ estimation and insufficient sample sizes. Some concerns were identified in eight studies (30.8%), which, despite lacking a true reference standard, utilized well-known oximeters like Masimo or employed large, validated datasets to mitigate the risk of bias.

In the flow and timing domain, D4, which addresses the range of SpO_2_ saturation intervals tested, including hypoxic conditions, the analysis revealed varied approaches across the studies. Although most studies (76.9%) involved healthy subjects, studies including non-healthy patients [46,48,49,56,60,61] applied various methods for SpO_2_ estimation. For example, Ghazal et al. [48] measured SpO_2_ in critically ill children using standard clinical oximeters, reporting ranges between 84 and 100%. Venkat et al. [49] employed Reflectance DAQ, Nellcor, and Masimo Radical 7 for pulmonary patients, with saturation levels ranging from 80 to 100%. Similarly, studies involving critically ill adults in the BIDMC dataset [56,60] reported SpO_2_ levels from 84 to 100%. The MIMIC database studies [46,61] included ICU patients but did not always specify exact saturation ranges. For healthy adults, hypoxia was artificially induced to achieve lower saturation ranges, as seen in Ogawa et al. [45] (76–98%), Priem et al. [23] (70–100%), Vijayarangan et al. [50] (60–100%), and Badiola-Aguirregomezcorta et al. [43] (70–100%).

Therefore, a low risk of bias was identified in four studies (15.4%) that tested broader SpO_2_ saturation intervals [23,43,45,50]. Conversely, a high risk of bias was found in 12 studies (46.2%), while some concerns were noted in 10 studies (38.5%), primarily due to reported saturation intervals that did not meet the FDA-recommended values.

Based on this quality assessment, several key areas require attention in future studies. Firstly, improving patient selection is critical. Studies should ensure a comprehensive demographic representation (age, gender, number of participants, and objective skin color classification) to mitigate selection bias and enhance the generalizability of findings. Secondly, enhancing reference standards is essential. Employing accurate and reliable ground truth methods for SpO_2_ estimation, such as arterial BGA, will improve the validation of AI models. Detailed reporting of reference standards is necessary to enable meaningful comparisons across studies.

Thirdly, optimizing flow and timing is important. Studies should clearly document the timing of tests and reduce delays between the index test and reference standard to ensure accurate measurements. Expanding the range of SpO_2_ saturation intervals, particularly under hypoxic conditions, is necessary to comprehensively evaluate the performance of models. Additionally, standardizing methodologies is crucial. Developing and adhering to standardized protocols for preprocessing techniques, feature extraction, and performance metrics reporting will enhance study comparability. Furthermore, using larger and more diverse datasets to train and validate AI models will reduce the risk of overfitting and improve generalizability.

Finally, ethical considerations must be addressed. Ensuring the ethical simulation of hypoxic conditions is necessary to balance the need for comprehensive testing with the safety of participants. By addressing these key recommendations, future research can improve the reliability, accuracy, and clinical applicability of AI-driven SpO_2_ estimation from PPG signals.

### 3.7. Study Limitations and Future Research Perspectives

This study has several limitations that should be considered for proper interpretation of the results and to guide future research in SpO_2_ estimation from PPG signals using AI. While AI models have shown promising results in controlled environments, the majority have not yet been validated in real clinical settings. Validation across diverse clinical and demographic conditions is crucial to ensure effectiveness in practice. Including longitudinal follow-up studies would also verify the consistency and stability of the models over time.

One major limitation is the variability in databases and oximeter types used across studies, which affects the comparability and generalizability of results. While large databases like MIMIC and BIDMC provide advantages due to their size, many smaller studies with fewer than 20 participants lack representativeness [25,43,57]. Developing standardized protocols for data collection, preprocessing, feature extraction, and model evaluation is essential to improve comparability between studies.

A key challenge in SpO_2_ estimation is the reliance on simulated data or artificially induced hypoxic conditions in healthy subjects. Although useful for testing, these methods do not fully replicate real clinical conditions where patients present with complex medical issues. Furthermore, inducing hypoxia requires specific clinical trial approval and presents ethical and safety concerns, limiting data availability for model training and evaluation. Techniques like administering supplemental oxygen or inducing physical exertion are used to simulate hypoxia [69], but these may introduce motion artifacts that affect PPG signal quality, especially in high-risk environments like ICUs or operating rooms [70]. Future studies should expand datasets to include more diverse patient populations, addressing these limitations.

Some AI models exhibit overfitting, showing significant performance differences between validation and test datasets. This suggests that models are overly tailored to training data and do not generalize well to unseen data. The lack of large and diverse datasets, as highlighted in the earlier analysis of studies with fewer participants, limits the models’ ability to learn representative features of the general population. Rigorous evaluation of model accuracy using invasive methods such as blood gas analysis as a reference is necessary. Standardized evaluation metrics should be developed, and exhaustive comparisons between different models should be made using these metrics.

Implementing cross-validation techniques like Leave One Subject Out (LOSO) is recommended to enhance model generalizability, as it ensures testing on entirely new subjects and accounts for inter-subject variability [82,83].

Promoting the publication of source code and using open platforms will improve transparency and reproducibility. Shared code and data enable other researchers to replicate results and validate findings, fostering collaboration and progress in the field.

The ethical challenges of simulating hypoxic conditions, combined with the need for robust validation, highlight the importance of improving reference standards and data collection protocols. Future studies must address these challenges while ensuring compliance with regulations like GDPR, protecting participant privacy, and fostering transparency to strengthen public trust in research.

Integrating emerging technologies, such as explainable AI (XAI), can enhance the interpretability of AI models in clinical settings [59]. Additionally, hybrid approaches that combine AI techniques with traditional physiological models may offer further improvements in SpO_2_ estimation accuracy and robustness, advancing patient care.

## 4. Conclusions

This work underscores the cutting-edge application of artificial intelligence in estimating SpO_2_ from PPG signals, highlighting significant technological advancements and methodological diversity. The AI models were systematically categorized, showcasing strengths across statistical approaches and advanced techniques like CNNs and ensemble methods. Comparing performance metrics such as Mean Absolute Error (MAE) and Root Mean Square Error (RMSE) across AI models is essential for improving SpO_2_ estimation reliability in clinical and controlled environments.

The results demonstrated significant diversity in AI methodologies, with Linear Regressors, Neural Networks, Ensemble Methods, and Support Vector Machines being frequently used algorithms, and Gaussian Process models showing high precision. Key features like statistical properties, waveform analysis, and time-domain representations of the signal illustrate the complexity of SpO_2_ estimation. The use of various software tools, including HeartPy for signal processing and TensorFlow/PyTorch for AI model implementation, along with open-source platforms, emphasizes a commitment to transparency and replicability in research.

Additionally, AI techniques integrated with remote photoplethysmography and cloud-based analytics with real-time notifications highlight the potential to enhance patient care, enabling user-friendly applications in devices like smartphones and smartwatches. This evolution points to the growing potential of AI-powered solutions beyond clinical settings, making SpO_2_ monitoring more accessible for daily use.

However, a key challenge remains in the reliance on non-invasive methods. The QUADAS-2 analysis revealed that most studies exhibited high or moderate risk of bias, primarily due to inadequate patient selection, limited SpO_2_ intervals, and the lack of reliable reference standards, which underscores the need for standardized methodologies and robust datasets to enhance the accuracy and generalizability of AI-driven SpO_2_ estimation. Additionally, combining AI with traditional physiological models can further enhance SpO_2_ accuracy and reliability.

In summary, this work spotlights technological advancements and interdisciplinary approaches that pave the way for improved non-invasive blood oxygen monitoring. Future research should prioritize the development of robust training datasets, refined feature selection, and model validation against accurate standards to boost generalization and clinical applicability. These steps will ensure that AI-based solutions become more robust, reliable, and clinically relevant across diverse healthcare environments.

## Figures and Tables

**Figure 1 bioengineering-11-01061-f001:**
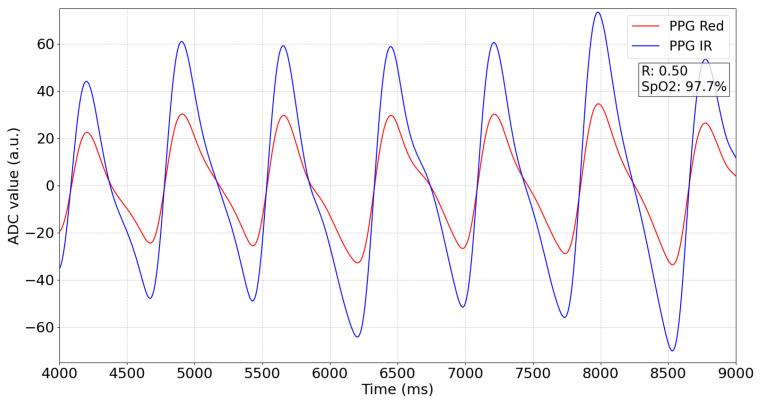
PPG signals captured at two wavelengths (red and infrared). The ratio of ratios (R) is calculated from the red (PPG Red) and infrared (PPG IR) signals. In this example, an R value of 0.50 results in an estimated SpO_2_ of 97.7%.

**Figure 2 bioengineering-11-01061-f002:**
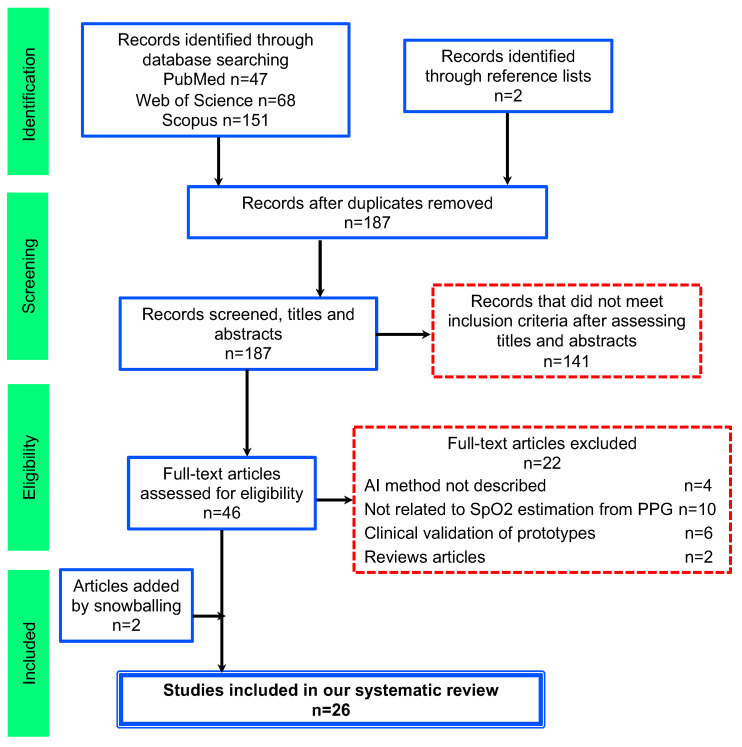
Flowchart following the PRISMA guidelines for systematic reviews.

**Figure 3 bioengineering-11-01061-f003:**
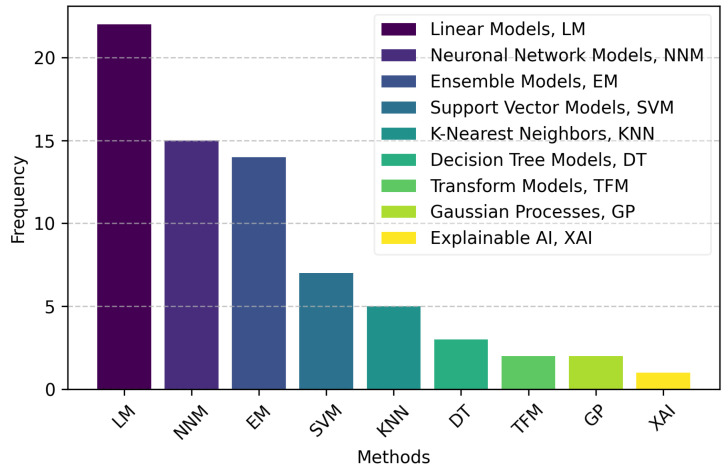
Overview of AI models used for SpO_2_ estimation, classified into categories based on scikit-learn’s framework of supervised learning models.

**Figure 4 bioengineering-11-01061-f004:**
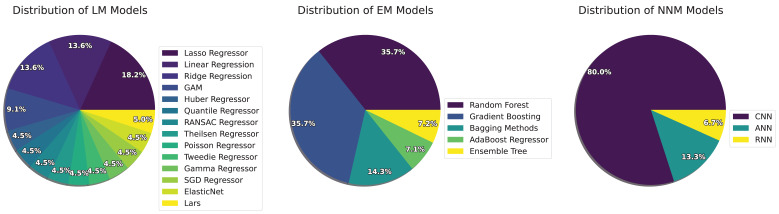
Distribution of AI models used in SpO_2_ estimation across three categories: Linear Models (LMs), Ensemble Models (EMs), and Neural Network Models (NNMs).

**Figure 5 bioengineering-11-01061-f005:**
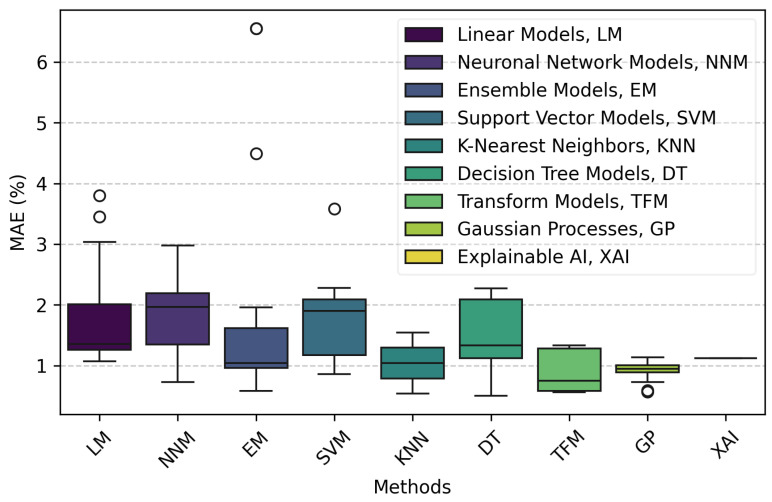
Boxplot illustrating the distribution of Mean Absolute Error (MAE) percentages across various AI model categories used for SpO_2_ estimation from PPG signals.

**Figure 6 bioengineering-11-01061-f006:**
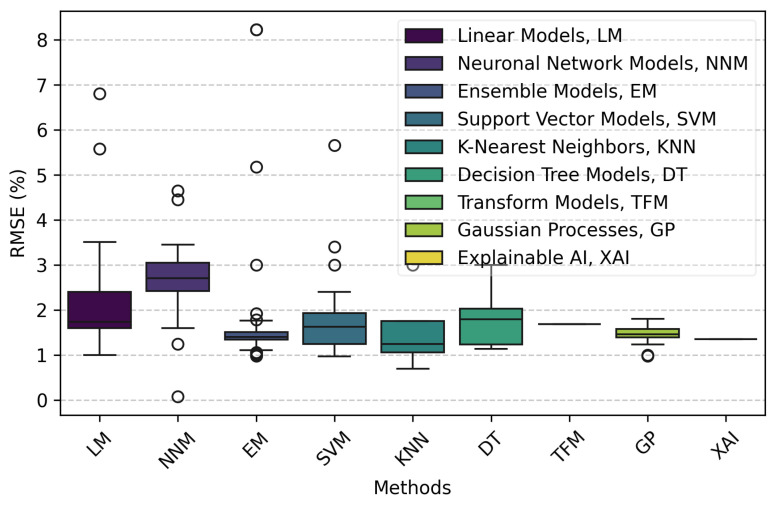
Boxplot illustrating the distribution of Root Mean Square Error (RMSE) percentages across various AI model categories used for SpO_2_ estimation from PPG signals.

**Figure 7 bioengineering-11-01061-f007:**
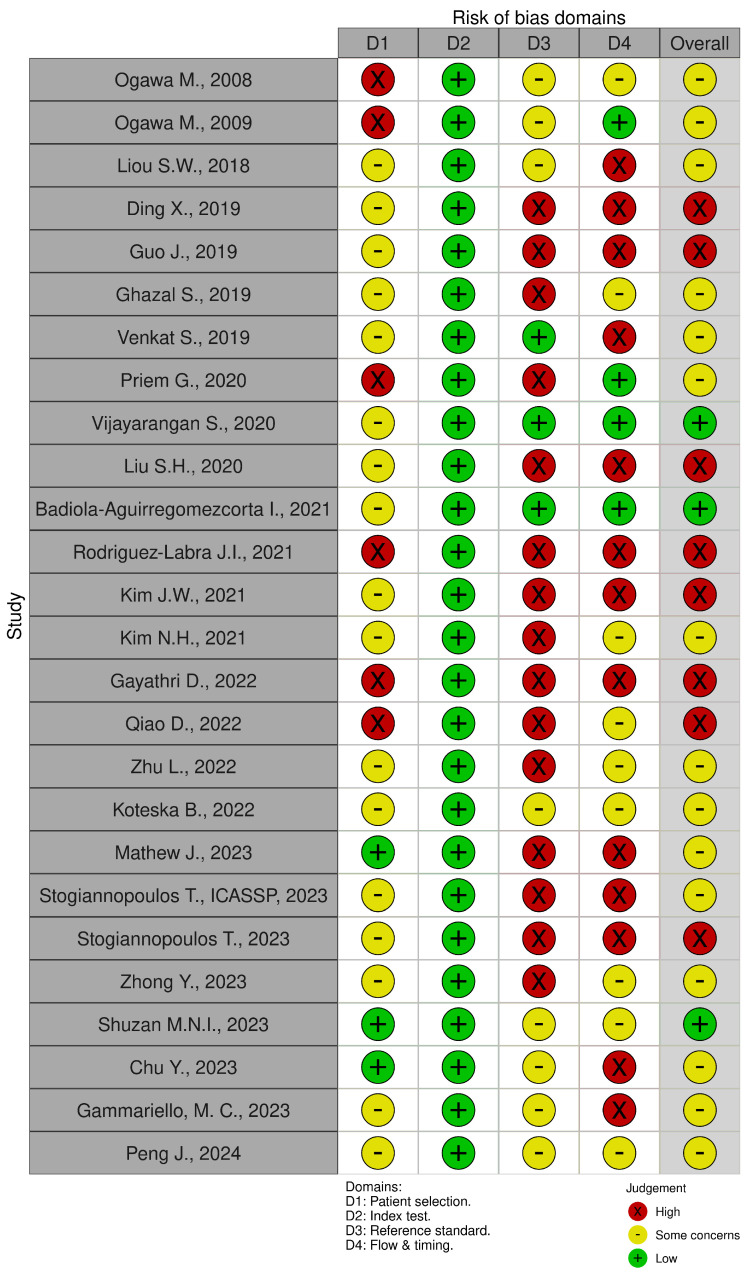
Risk of bias assessment in individual studies judged according to QUADAS2 [22,23,25,31,42,43,44,45,46,47,48,49,50,51,52,53,54,55,56,57,58,59,60,61,62,63].

**Table 1 bioengineering-11-01061-t001:** Search strings, keywords and items per database.

Indexing Terms	Items (n)
**Web of Science**	
#1 “oximet*” OR “oxygen saturation” OR “SpO_2_”	46,561
#2 “photoplethysmography” OR PPG	15,740
#3 “Artificial Intelligence” OR “AI” OR “Machine Learning” OR “Deep Learning” OR “Neural Networks” OR “Support Vector Machines”	
OR “Genetic Algorithms” OR “supervised learning” OR “unsupervised learning”	1,246,213
#4 “accuracy” OR “precision” OR “measurement accuracy” OR “error” OR “bias” OR “reliability”	3,322,177
#5 #1 AND #2 AND #3 AND #4	68
**PubMed**	
#1 “oximet*” OR “oxygen saturation” OR “SpO_2_”	60,310
#2 “photoplethysmography” OR “PPG”	8764
#3 “Artificial Intelligence” OR “AI” OR “Machine Learning” OR “Deep Learning” OR “Neural Networks” OR “Support Vector Machines”	
OR “Genetic Algorithms” OR “supervised learning” OR “unsupervised learning”	1,246,213
#4 “accuracy” OR “precision” OR “measurement accuracy” OR “error” OR “bias” OR “reliability”	1,419,845
#5 #1 AND #2 AND #3 AND #4	47
**Scopus**	
#1 “oximet*” OR “oxygen saturation” OR “SpO_2_”	126,043
#2 “photoplethysmography” OR “PPG”	17,684
#3 “Artificial Intelligence” OR “AI” OR “Machine Learning” OR “Deep Learning” OR “Neural Networks” OR “Support Vector Machines”	
OR “Genetic Algorithms” OR “supervised learning” OR “unsupervised learning”	771,104
#4 “accuracy” OR “precision” OR “measurement accuracy” OR “error” OR “bias” OR “reliability”	6,992,893
#5 #1 AND #2 AND #3 AND #4	151

**Table 2 bioengineering-11-01061-t002:** Main information about selected articles.

Description	Results
Timespan	2008–2024
Sources (journals, books, etc.)	22
Documents	26
Average citations per document	11.64
Average citations per year per doc	4.14
References	797
Articles	17
Conference papers	11
Keywords	391
Authors	131
Single-authored documents	0
Documents per author	0.23

**Table 3 bioengineering-11-01061-t003:** Selected studies from the search query sorted by year of publication. First author’s name, year, database or type of oximeter, and SpO_2_ range are shown.

Reference	Database/Type of Oximeter	SpO_2_ Range
Ogawa M., 2008 [44]	Masimo Radical with LNOP DCI sensor	83–100%
Ogawa M., 2009 [45]	Masimo Radical with LNOP DCI sensor	76–98%
Liou S.W., 2018 [46]	MIMIC Database	–
Ding X., 2019 [42]	Nellcor PM10N and iPhone 7 Plus	–
Guo J., 2019 [47]	Wearable smartband system	–
Ghazal S., 2019 [48]	ERB Database 2016–1210, 4061	84–100%
Venkat S., 2019 [49]	Reflectance DAQ, Nellcor, Masimo Radical 7/BGA test by GEM Premier 3000	80–100%
Priem G., 2020 [23]	Wrist-worn reflectance pulse oximeter	70–100%
Vijayarangan S., 2020 [50]	Reflectance DAQ, Nellcor, Masimo Radical 7/BGA test by GEM Premier 3000	60–100%
Liu S.H., 2020 [22]	Reflectance Forehead MAX30102/Finger Rossmax SA310	–
Badiola-Aguirregomezcorta I., 2021 [43]	Reflective ear pulse oximeter LAVIMO/BGA test	70–100%
Rodriguez-Labra J.I., 2021 [51]	A wearable multi-channel PPG prototype	–
Kim J.W., 2021 [52]	58 PPG datasets	–
Kim N.H., 2021 [25]	Lab-made rPPG/CMS-50E pulse oximeter	90–98%
Gayathri D., 2022 [53]	Raspberry Pi-4 board	–
Qiao D., 2022 [54]	PURE Dataset/Face rPPG/CMS-50E pulse oximeter	90–100%
Zhu L., 2022 [55]	Finger EMAY EMO-80 pulse oximeter	94–99%
Koteska B., 2022 [56]	BIDMC Dataset	85–100%
Mathew J., 2023 [57]	Self-collected dataset/CMS-50E pulse oximeter	–
Stogiannopoulos T., 2023 [58]	JPD-500D ControlBios Oxicore	–
Stogiannopoulos T., 2023 [31]	JPD-500D ControlBios Oxicore	–
Zhong Y., 2023 [59]	Smart wearable SleepEaze with MAX30105 sensor/CMS-50E pulse oximeter	90–100%
Shuzan M.N.I., 2023 [60]	BIDMC dataset	84–100%
Chu Y., 2023 [61]	MIMIC III dataset	–
Gammariello M.C., 2023 [62]	MTHS dataset/M70 pulse oximeter	–
Peng J., 2024 [63]	UBFC Dataset/CMS-50E pulse oximeter	93–100%

## Data Availability

The data supporting the findings of this study are available upon request from the corresponding author.

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
