# Peer review of "Evaluating AI Methods for Pulse Oximetry: Performance, Clinical Accuracy, and Comprehensive Bias Analysis"

_bioengineering, 2024, doi:10.3390/bioengineering11111061_

Round 1
Reviewer 1 Report
Comments and Suggestions for Authors
Overall, the authors had conducted a comprehensive and systematic review.
- There is only one key comment regarding the title of manuscript: the title is not clear. It does not specify which aspects of the applications of pulse oximetry would be.
Reviewer 2 Report
Comments and Suggestions for Authors
The paper calls: "Assessing the Reliability of AI in Pulse Oximetry: Performance Evaluation and Comprehensive Bias Analysis" and it is large review article concerned of Pulse Oximetry measurement topic. The advantage of article large quantity of data with long references list 71 ones.
The main disadvantage of article is absence of technical details (just links in first paragraph in introduction). In my opinion authors should add pictures and text descriptions which describes methods of oximetry in details.
